# Preparation of Puerarin Chitosan Oral Nanoparticles by Ionic Gelation Method and Its Related Kinetics

**DOI:** 10.3390/pharmaceutics12030216

**Published:** 2020-03-02

**Authors:** Jie Yan, Zhi-Yu Guan, Wei-Feng Zhu, Ling-Yun Zhong, Zhuo-Qi Qiu, Peng-Fei Yue, Wen-Ting Wu, Jing Liu, Xiao Huang

**Affiliations:** School of Pharmacy, Jiangxi University of Traditional Chinese Medicine, Nanchang 330004, China; jzy0624513@163.com (J.Y.); q2651776723@126.com (Z.-Q.Q.); ypfpharm@126.com (P.-F.Y.); wuwenting0109@163.com (W.-T.W.); liujing860828@163.com (J.L.); huangxiaosky@sina.com (X.H.)

**Keywords:** puerarin, nanoparticles, Box–Behnken, pharmacokinetic, LC-MS/MS, rats

## Abstract

In this paper, as an active ingredient, puerarin chitosan nanoparticles (Pur-CS/TPP-NPs) are prepared by an ionic gelation method. The chitosan (CS) concentration, pH of the CS solution, sodium tripolyphosphate (TPP) concentration, stirring speed, stirring time, ultrasonic power, and dosage are used as single factors for investigation, and the encapsulation efficiency, drug loading capacity, particle size, and polydispersity index (PDI) are used as indicators for investigation. The optimal prescription is determined using the Box–Behnken effect surface design method. The characterization of the best formulation, which is determined via an in vitro release assay and liquid chromatography/tandem mass spectrometry (LC-MS/MS) analysis methods, is used here for pharmacokinetic studies. An in situ single-pass intestinal perfusion model is used to investigate drug absorption in the intestine. After characterization, the morphologies of the nanoparticles are intact. It can be seen from the in vitro release experiments that the equation fitted by the nanoparticles is the Higuchi model, the nanoparticle release process is very stable and without sudden release, indicating that the nanoparticles are well-released in vitro. The pharmacokinetic results and the in situ single-pass intestinal perfusion model study show that the bioavailability and absorption of Pur-CS/TPP-NPs were significantly higher than Pur. Thus, all the results show that the prepared nanoparticles can significantly improve the bioavailability of Pur, and we hope to lay the foundation for the development of new products of Pur.

## 1. Introduction

Puerarin (Pur) is the main active ingredient of the leguminous plant *Pueraria lobata*. It has a variety of biological functions and is widely used in various clinical fields. The ingredient can relieve coronary heart disease angina pectoris, provide adjuvant treatment for myocardial [1,2] and cerebral infarction, and promote the recovery of neurological functions [3,4,5]. However, Pur is a BCS IV drug with low oral bioavailability [6]. Oral Pur is mostly excreted in its original form through feces, and only a small part is absorbed into the blood and distributed to various tissues and organs [7,8]. At present, there are only injections and eye drops in clinical use, and the injections require long-term use, and also have poor compliance and are prone to intravascular hemolysis, allergic asthma, anaphylactic shock, fevers, and other adverse reactions [9,10].

Oral administration is the most commonly used and ideal choice of drug administration route due to its suitability, ease of administration, and portability; thus, new oral drug delivery systems have become a research hot spot [11,12]. However, many of the bioactive ingredients of traditional Chinese medicine (TCM) are poorly soluble, which leads to low oral bioavailability and delivery problems. This makes the efficacies of TCM decrease or administration doses increase [13,14,15]. Consequently, oral formulations with improved absorption of Pur have attracted widespread attention. In recent years, a series of studies on improving oral bioavailability have been reported, including the use of nanotechnology.

Nano-scale drug delivery systems have attracted great attention due to their unique biological properties. This area has become a bright field of research [15]. Nanoparticles (NPs), as a new generation of submicron-sized drug delivery systems, have the advantages of controlling drug release, avoiding drug degradation or leakage, good targeting, and improving bioavailability, among other advantages. Nanoparticle-based drug delivery systems are a research hotspot for new drug delivery technologies and new dosage forms [16].

In this paper, we use water-soluble chitosan (CS) to prepare Pur nanoparticles through the ionic crosslinking of CS and sodium tripolyphosphate (TPP). This method does not require the addition of any harmful reagents, the reaction conditions are mild, and the method is relatively safe, with relatively low cost [17,18]. Compared with ordinary CS, the water-soluble CS used in this experiment has many advantages. It can be dissolved without an acid and was dissolved with ultrapure water during the experiment and the preparation process of nanoparticles is simple. Finally, the material expands the drug application range of the nanoparticle preparation method via ion gelation. Unstable drugs in acid can also be prepared by this method, dealing with the restrictions of many drugs, and opening areas for possible new drugs to be used in delivery systems.

The ionic gelation method consists of the spontaneous reaction of cationic CS with an anionic crosslinking agent, usually TPP, forming a polyelectrolyte complex named TPP/CS. This complex is stabilized by the cross-linked electrostatic interaction between CS-NH_3_^+^ and TPP-O- groups, resulting in a three-dimensional entanglement that precipitates from an aqueous solution in the form of gel-like nanoparticles, also named microgels, where encapsulation then occurs, causing Pur to form nanoparticles [19,20]. This method is simple to operate, and it is easily adaptable to industrialized production. The prepared nanoparticles have a uniform particle size and complete morphology [21,22].

Therefore, in this paper, Pur oral nanoparticles are prepared by an ionic gelation method. The prepared nanoparticles are optimized for the detailed formulation process, and characterization, pharmacokinetics, and intestinal in vivo intestinal absorption kinetics are analyzed herein. The aim of the present study is to lay the foundation for future Pur-based oral drug development, clinical application, and industrial development. The researchers also hope that this method will provide future research directions for BCS IV drugs, especially those that are unstable in acids.

## 2. Materials and Method

### 2.1. Materials and Instrumental Reagents

Puerarin (purity over 99%) was purchased from Chengdu Pfeide Biotechnology Co., Ltd. (Chengdu, Sichuan. China). CS was made in the laboratory, with a molecular weight between 20,000 to 30,000 and >99% deacetylation. Sodium tripolyphosphate was purchased from the Shanghai Yien Chemical Technology Co., Ltd. (Beijing, China). Trypsin and pepsin were purchased from Beijing Solarbio Science and Technology Co., Ltd. (Beijing, China). Acetonitrile, methanol, and formic acid were MS-grade reagents purchased from Tedia Company, Inc. Deionized water was purified via a Millipore Milli-Q system (Millipore, Bedford, MA, USA). The other chemical reagents were of an analytical grade.

### 2.2. Animals

Adult male Sprague–Dawley rats (SD, 180~220 g) were obtained from the Animal Experiment Technology Center of the Jiangxi University of Traditional Chinese Medicine (Jiangxi, China) and housed in a temperature and humidity-controlled room at 24 ± 2 °C and 50 ± 10%, respectively. Water and food were available ad libitum. The rats were fed for two weeks and fasted for 12 h before treatment. The whole experimental protocol was approved by the Institutional Animal Care and Use Committee at the Jiangxi University of Traditional Chinese Medicine. Laboratory animal license number is SCXK (Gan) 2018-0003. (25 January.2018).

### 2.3. Experimental Method

#### 2.3.1. Establishment of Analytical Method for Determination of Pur-CS/TPP-NPs

The chromatographic column used in the experiment was an Agilent TC-C18 column (250 mm × 4.6 mm, 5 μm), with a mobile phase of methanol and water (30:70), a volume flow rate of 1 mL/min, an injection volume of 10 μL, and a column temperature of 30 °C. The specificity of the analytical methods was good, and the linear relationship was 0.012 to 0.072 mg/mL, showing a good linear relationship. The precision, stability, repeatability, and RSD of the sample recovery rates were 0.47%, 0.33%, 0.37%, 1.5%, respectively, showing that the established analytical method was feasible.

#### 2.3.2. Preparation of Pur CS Nanoparticles by Ion Gelation

Sodium TPP and CS were separately weighed to 1000 mg in a 10 mL volumetric flask, diluted to volume with water, and then left to dissolve in the refrigerator for 6 h or more. Next, 10 mg of Pur was ultrasonically dissolved in 5 mL of a 5% ethanol aqueous solution, and 100 mg of laboratory-made CS was dissolved in water to make a 2.0 mg/mL solution. The pH of CS solution was adjusted to 4.5 with 1% dilute hydrochloric acid and saturated sodium hydroxide. Then, 6 mL of the CS solution was transferred into a beaker and placed under a magnetic stirrer. Next, 5 mL of the Pur ethanol solution was slowly dripped into the CS solution, then added to the mixture at 500 r/min for 30 min. After the stirring was completed, 1 mL of a 0.6 mg/mL TPP solution was accurately weighed and slowly dropped into a mixed solution of Pur and CS. After all of the additions, stirring was continued for 30 min at a speed of 500 r/min until the system showed significant opalescence.

#### 2.3.3. Encapsulation Rate and Drug Loading Are Determined

Precisely took 200 μL of Pur-CS/TPP-NPs suspension and placed it in the microporous membrane. After centrifugation at 12,000 r/min for 30 min, the retentate was taken and filtered through a 0.22 μm filter. Precisely took 100 μL of the filtrate, made up to 10 mL of methanol, mixed well, and measured according to the above chromatographic conditions.
(1)$$\mathrm{EE}\%=\frac{C-C1}{C}\times 100\%$$
(2)$$\mathrm{DL}\%=\frac{W1}{W1+W2}\times 100\%$$
where C is the concentration of total *Pur* in the suspension; C1 is the concentration of unencapsulated drug in the suspension; W1 is the quality of the encapsulated drug; W2 is the quality of all excipients.

#### 2.3.4. Single Factor Investigation

Based on the above stock solutions, the CS concentration was 0.5, 1.0, 1.5, 2.0, 2.5, 3.0 mg/mL, respectively. The pH of the CS solution was 4.0, 4.5, 5.0, 5.5, 6.0, 6.5, the TPP concentration was 0.5, 1.0, 1.5, 2.0, 2.5 mg/mL, stirring speed was 100, 300, 500, 700, 900 r/min, after adding TPP solution, the stirring time was 15, 30, 60, 90, 120 min, the temperature was room temperature, 25 °C, 30 °C, 35 °C, 40 °C, and the effect on the particle size, PDI, encapsulation efficiency and drug loading of Pur-CS/TPP-NPs was measured.

#### 2.3.5. Box–Behnken Effect Surface Method to Optimize Prescription

According to the single factor test results, using the Box–Behnken response surface optimizes the preparation process. For the prepared Pur-CS/TPP-NPs, factor A (CS concentration mg/mL), B (pH of CS solution), C (TPP concentration mg/mL), D (speed r/min) were selected. The experimental design is shown in Table 1.

#### 2.3.6. Optimal Prescription Verification Test

According to the Box–Behnken effect surface optimization experiment, the best prescription was obtained, and the best prescription was verified in three parallel experiments.

#### 2.3.7. Pur-CS/TPP-NPs Morphological Characterization

We took a proper amount of the Pur-CS/TPP-NPs suspension, dropped it onto the slide, and slowly covered the coverslip to prevent air bubbles. The microscope was adjusted so that the polarization angle was 90°, and the field of view was dark black. The sample slide was placed under an electron microscope for observation. Next, 10 μL of the diluted sample was dropped on a copper mesh covered with a support film, which was allowed to stand for 5 min, then blotted dry with filter paper. Then, 10 μL of a 0.5% phosphotungstic acid solution was added to the copper mesh for 5 min, then placed under a transmission electron microscope for observation.

#### 2.3.8. In Vitro Drug Release Experiment

The in vitro release behavior of nanoparticles was determined by a dialysis method, using artificial gastric juice and artificial intestinal juice as release media, respectively. We precisely measured the nanoparticle suspension, and 2 mL of the Pur solution was transferred into a dialysis bag (retaining a relative molecular weight of 3500), both ends were fastened, and the bag was placed in a beaker containing 150 mL of the release medium. The test was carried at a temperature of 37 ± 0.5 °C and a 100 r/min stirring rate. We sampled 1 mL at 0.25, 0.5, 0.75, 1, 2, 3, 4, 6, 8, 10, 12, and 24 h, respectively, while replenishing the same amount of medium at the same temperature. The drug concentration and the cumulative release rate were calculated by filtration through a 0.22 μm microporous membrane [23,24]. The cumulative release percentage Qn% of the drug at various time points was calculated by the following formula:(3)$$\mathrm{Qn}\%=(\mathrm{CnV}+\overset{i=n-1}{\underset{i=1}{\sum }}\mathrm{CiVi})/M$$
where Qn% is the cumulative release percentage of the drug at the nth time point, Cn is the concentration of Pur at the nth sampling, V is the total volume of the release medium, Vi is the sampling volume, and the quality of Pur in the dialysis bag is M.

#### 2.3.9. Bioavailability Studies

##### Administration and Sampling

Twenty-seven SD rats were randomly divided into three groups, with 9 rats in each group. The Pur group had Pur 200 mg/kg administered by intragastric administration. The nanoparticle group underwent the intragastric administration of Pur-CS/TPP-NPs (equivalent to Pur 200 mg/kg). The rats were fasted before the experiment and were free to drink water. After administration, 0.3 mL of blood was collected from the orbital vein at 15 min, 30 min, 45 min, 1, 1.5, 2, 4, 6, 8, 12, and 24 h, and heparin was anticoagulated. The blood was centrifuged at 4000 r/min for 10 min in a refrigerated centrifuge, and the supernatant was separated and stored in a refrigerator.

##### Whole Blood Sample Preparation

Precisely took 100 μL of plasma, added 200 μL of chromatographically pure methanol, 100 μL of the internal standard, and mixed the solution in a vortex with a vortex mixer for 3 min. Then, the protein was precipitated by centrifugation at 15,000 r/min for 10 min in a refrigerated centrifuge, and the supernatant was transferred to another centrifuge tube, then dried at 45 °C under a nitrogen atmosphere. The residue was dissolved in 100 μL of methanol, vortexed for 5 min, centrifuged at 15,000 r/min for 10 min, and the supernatant was then taken for liquid chromatography/tandem mass spectrometry (LC-MS/MS) analysis.

##### Pur Whole Blood Sample Determination

Pur in the whole blood was quantified by LC-MS/MS using a liquid chromatography system (SHIMADZU, Japan) coupled to a 4500 Q TRAP quadrupole linear ion trap hybrid mass spectrometer (Applied Biosystems, MDS Sciex, Foster City, CA). All chromatographic separations were performed with an Agilent ZORBAX 300SB-C18 column (150 mm × 2.1 mm, 5 µm). The mobile phase consisted of acetonitrile and 0.3% formic acid water at a total flow rate of 0.35 mL/min. An electrospray ionization (ESI) source was applied and operated in the positive ion mode. Multiple reaction monitoring (MRM) was carried out with transitions of m/z of 417.0/297.2, and m/z values of 370.2/252.0 were used to quantify Pur and the internal standard (lansoprazole), respectively. The specificity of the analytical methods was examined separately, and the linear relationship was determined to be a good linear relationship between 5–120 nG/ml. The relative standard deviation (RSD) values of precision, stability, repeatability, and sample recovery were 2.54%, 4.23%, 1.21%, 1.32%, respectively, showing that the established analytical method was feasible.

##### Pharmacokinetic Data Analysis

The peak plasma concentrations (C_max_) and the time of their occurrence (T_max_) were noted directly from the individual whole blood concentration versus time profiles. The area under the whole blood concentration–time curve (AUC_0–t_) was estimated by the linear trapezoidal method.

##### Statistical Analysis

Results are reported here as mean ± standard deviation (S.D.). Statistically significant differences were determined by ANOVA, followed by Tukey’s test for multiple comparisons at a significance level of *p* = 0.05.

#### 2.3.10. Rats In Situ Single-Pass Intestinal Perfusion Model Studies

Rats were fasted overnight (12–18 h) with free access to water and were allocated to different experimental groups at random before the experiments. The surgical procedures for the in situ single-pass intestinal perfusion (SPIP) were performed as previously described. Briefly, rats were anesthetized with an intraperitoneal injection of chloral hydrate at a dose of 1.5 g/kg and were placed under an infrared light to maintain a body temperature of 37 °C. The abdomen was opened with a midline incision of 3–4 cm. The duodenum (10 cm down from 1 cm from the pylorus), jejunal (10 cm down from 1 cm of the duodenum), ileum (10 cm down from the upper end of the cecum), colon (10 cm from the back end of the cecum), and 10 cm for each intestine were cannulated on two ends with flexible PVC tubing. The exposed segment was covered with a cotton pad soaked in a 37 °C normal saline solution. The isolated segment was rinsed with isotonic saline (37 °C) to clean out any residual debris [25,26].

The Pur and Pur-CS/TPP-NPs were separately dissolved in the K-R solution; the low, medium, and high concentrations were 40, 80, and 160 μg/mL, respectively; and the pH of the K-R solution was adjusted to 7.4. At the start of the study, the perfusion solution was perfused through the intestinal segment at a flow rate of 0.2 mL/min for 30 min using a peristaltic pump. Following reaching a steady-state, the intestinal perfusate samples were collected at 15 min intervals for a duration of 2 h (15, 30, 45, 60, 75, 90, 105, 120 min) in 5 mL glass vials. All samples, including the perfusion samples from both the inlet and outlet drug solutions at different time points, were immediately assayed by HPLC. All glass vials were weighed before and after the perfusion. At the end of the experiment, the lengths of the perfused intestinal segments were accurately measured

The effective permeability (*P_eff_*) and absorption rate constant (*K_a_*) of the drug was calculated using the following equation:(4)$$P_{eff}=\frac{-Q_{in}ln\left(\frac{C_{out}Q_{out}}{C_{in}Q_{in}}\right)}{2\pi rl}$$
(5)$$K_{a}=\left(1-\frac{C_{out}Q_{out}}{C_{in}Q_{in}}\right)\frac{Q}{V}$$
(6)$$V={\pi r}^{2}l$$
where *P_eff_* is the effective permeability coefficient (cm/min), *K_a_* is the absorption rate constant, *Q_in_* is the flow rate (mL/min) of the inlet solution, *Q_out_* is the measured perfusate exit flow (mL/min) for the specified time interval using the actual intestinal perfusate density (g/mL), *Q* is the flow rate (mL/min) of the pump, *C_in_* is-the concentration of the perfusate flowing in (µg/mL), *C_out_* is the concentration of the perfusate flowing out, *V* is the volume of the perfused segment (mL), l is the length of the intestinal segment (cm), and *r* is the radius of the intestinal segment (cm).

## 3. Results and discussion

### 3.1. Preparation of Pur-CS/TPP-NPs

#### 3.1.1. Single Factor Investigation

Through single factor investigation, when the CS concentration was 0.5–2.0 mg/mL, the encapsulation efficiency and drug loading of the nanoparticles were best, and the particle size of the nanoparticles was about 150 nm, the polydispersity index (PDI) was stable, and the solution was clarified. When the concentration of CS exceeded 2 mg/mL, the particle size of the nanoparticles was very large, and the suspension had obvious floc. When the pH of the CS solution was 4.0–4.5, the particle size, PDI, encapsulation efficiency and drug loading of the nanoparticles were the best. The concentration of TPP had a great influence on the particle size of nanoparticles. When the concentration of TPP was more than 1.5 mg/mL, the nanoparticle suspension had obvious floc and the particle size was larger than 1000 nm. When the concentration of TPP was 0.5–1.5 mg/mL, the particle size, PDI, encapsulation efficiency, and drug loading were the best. When the stirring speed was lower than 300 r/min or more than 700 r/min, the particle size of the nanoparticles was very large and there was a clear suspension. After the TPP solution was added dropwise, it needed to be stirred for a certain period of time until the nanoparticle suspension had obvious opalescence. When the stirring time was too long, the particle size of the nanoparticles became large, so the stirring time was 60 min. Increasing the temperature caused the particle size of the nanoparticles to increase, and the encapsulation efficiency and the drug loading amount were very low, so the temperature was set to room temperature. Based on the results of a single factor review, the pH of the CS solution, the concentration of CS, and the concentration of TPP were selected.

#### 3.1.2. Box–Behnken Effect Surface Method Optimization Results

According to the Design Expert 8.0.6 software, the average particle size, polydispersity coefficient (PDI), encapsulation efficiency, and drug loading amount were measured. Each index was set to a normalized value of 0–1, and each indicator was returned to the geometric mean of the value, and the total value of the normalized value (OD) was obtained. These were calculated according to the formula OD = (d_1_d_2_...d_k_)^1/k^, where k is the number of indicators. The larger the encapsulation efficiency and drug loading amount, the better the average particle size and the smaller the PDI value. The Hassan method was used to calculate the normalized values for d_max_ and d_min_ as in [27]. The Box–Behnken effect surface method experimental results are shown in Table 2.

The OD value of each factor variance analysis results is shown in Table 2.

d_max_ = (Y_i_ − Y_min_)/(Y_max_ − Y_min_)

d_min_ = (Y_max_ − Y_i_)/(Y_max_ − Y_min_)

Through the test results of various indicators F, regression analysis was performed using Design-Expert 8.0.6 software. The obtained regression equation OD = 0.880 + 0.059A + 0.056B − 0.130C + 0.024D − 0.075 AB − 0.058AC + 0.110AD + 0.110BC + 0.080BD − 0.24CD − 0.300A^2^ − 0.340B^2^ − 0.140C^2^ − 0.330D^2^.

It can be seen from the equation that the effect of the encapsulation efficiency of the nanoparticles on the selected range of factors is as follows: C > A > B > D, TPP concentration > CS concentration > pH of CS solution > speed. At the same time, various factors interact with each other. During the experiment, it was found that the concentration control of TPP was the biggest factor affecting the particle size and PDI of the nanoparticles. When the TPP concentration was too high, the particle size of the nanoparticles and the PDI was very large, indicating that the nanoparticles were not uniformly dispersed. This was consistent with the optimized experimental results. The analysis of variances in the regression model is shown in Table 3. 

The regression equation shows that R^2^ = 0.9286, where the *p*-value of the misfit test was 0.0993, meaning that there was no significant difference with respect to the Pur error. This shows that the regression model was successfully fitted, and can accurately express the relationship between the response value and various factors and that we could use this model to speculate and analyze the experimental results. From the results of the significance test, it was found that C (CS concentration and TPP concentration) was significantly higher in the model (*p* < 0.05). In the results of the interaction terms, AB, AC, AD, BC, and BD all had *p*-values > 0.05, indicating that there were no significant differences between the interaction terms. Here, CD = 0.0006, a significant difference, indicating that the TPP concentration and speed had a significant effect on the nanoparticles and that the simultaneous change in TPP concentration and rotation speed had a great impact on the nanoparticles. This is consistent with the results obtained during the experiment, where CS concentration and TPP concentration were two factors that affected the particle size, PDI, encapsulation efficiency, and drug loading. Additionally, predicting the concentration of TPP was the main factor that affected the crosslinking effect and promoted the formation of a more uniform spatial network structure. As the TPP concentration increased slightly, the particle size of the nanoparticles immediately increased. A large amount of TPP would form more intramolecular cross-links with chitosan, which makes the molecules more easily polymerized, having a larger particle size and uneven distribution. Too little TPP resulted in less chitosan formation and less intramolecular cross-linking, making it difficult for the molecules to polymerize and load drugs. Therefore, a quadratic multiple regression equation was used to fit experimental data, determining the level of one factor as the center point and predicting the relationship diagram with the other three factors, as well as predicting the optimal prescription ratio of Pur CS nanoparticles. The results are shown in Figure 1.

### 3.2. Optimal Prescription Verification Test

According to the results of the Box–Behnken effect surface experiment, the final optimization prescription was determined. Here, the CS concentration was 1.41 mg/mL, the pH of the CS solution was 4.45, the TPP concentration was 0.49 mg/mL, and the rotation speed was 558 r/min. For the convenience of experimental operation, the optimal prescription was determined to be a CS concentration of 1.40 mg/mL, pH of the CS solution of 4.5, TPP concentration of 0.50 mg/mL, and rotation speed of 560 r/min. Three experiments were performed in parallel according to the predicted prescription, and each evaluation index was close to the predicted value, indicating that the established regression equation was predictive, and the results of this are shown in Table 4. It is known from the results that after repeated experiments, the particle size of the nanoparticles was 126.289 ± 1.175, the PDI was 0.122 ± 0.0005, the encapsulation rate was 94.951 ± 0.468, and the drug loading was 39.596 ± 0.388. All values were less than 1% when compared with the predicted values. The results show that the reproducibility of the optimized prescription was good.

### 3.3. Pur-CS/TPP-NPs Morphological Characterization Analysis

It can be seen from the transmission electron micrograph results that from 100 nm to 150 nm under the ransmission electron microscope, the shape of the nanoparticles was round and that the particle size distribution was uniform. It is shown in Figure 2. The results show that the nanoparticles prepared with this optimized formulation have a uniform particle size distribution and uniform size.

### 3.4. In Vitro Drug Release Analysis

The cumulative release of Pur and Pur-CS/TPP-NPs in vitro was determined according to simulated artificial gastric juice and artificial intestinal juice. It can be seen from the above results that the cumulative release of the Pur bulk drug in artificial intestinal juice and artificial gastric juice for 8 h was 99%, and the subsequent release was close to steady. The Pur-CS/TPP-NPs continued to release in artificial intestinal juice and artificial gastric juice after 8 h. The fitted equation was the Higuchi model, indicating that the nanoparticles were released well in vitro. The results of the in vitro release show that the ionic gelation method formed by CS and TPP effectively encapsulates Pur, achieving a slow-release effect for Pur and effectively increasing the cumulative release of Pur. The results are shown in Figure 3 and Table 5 below:

### 3.5. Pharmacokinetic Analysis

The plasma concentration-time profiles following Pur and Pur-CS/TPP-NPs administration are shown in Table 5. The results are statistically significant for a *p*-value of less than 0.05, except when compared between Pur and Pur-CS/TPP-NPs. As can be seen from the pharmacokinetic results, the Pur group had the following properties: AUC_0–24_ (mg/L·h) of 250.087 ± 32.156, AUC_0–∞_ (mg/L·h) = 250.091 ± 28.398, T_max_ (h) = 0.500 ± 0.031, C_max_ (mg/L) = 191.830 ± 17.963. The Pur-CS/TPP-NP group had the following properties: AUC _0–24_ (mg/L·h) = 1108.583 ± 234.534, AUC_0–∞_ (mg/L·h) = 1182.926 ± 265.746, T_max_ (h) = 1.000 ± 0.076, C_max_ (mg/L) = 343.314 ± 56.760. This indicates that the nanoparticle group AUC_0–24_ (mg/L·h) value was 4.432 times that of the Pur group, and that the T_max_ changed during 0.5 h to 1 h, and the maximum release time of the nanoparticles shifted back, indicating that the nanoparticles had a sustained-release effect. Pur-CS/TPP-NPs increased by 443.279% relative to the Pur relative bioavailability. From the pharmacokinetic results, the Pur-CS/TPP-NPs effectively improved the bioavailability of Pur. This shows that CS and TPP effectively play a role in purifying Pur. It was the Pur package that promoted absorption and improved bioavailability. The results of this are shown in Figure 4 and Table 6 below.

### 3.6. Influence of Intestinal Segment on the Lymphatic Transport of Pur and Pur-CS/TPP-NPs

In the early stage, by examining the effects of the pH value of the perfusate, the perfusion speed, the peristaltic pump, etc., it was finally determined that the absorption was best when the perfusate pH was 7.4 and the flow rate was 2 mL/min. In in situ single-pass intestinal perfusion model studies, because the small intestine and colon not only absorb drugs but also absorb and secrete water, the volume of the perfusate changed, so the volume of the perfusate needed to be measured. At present, the phenol red and gravimetric methods are commonly used volume measurement methods. However, Pur could promote the absorption of phenol red in each intestine, so the phenol red method was not suitable for this experimental study. Therefore, the gravimetric method was used to determine the volume of perfusate in this test, and the intestinal absorption parameters *K_a_* and *P_eff_* were calculated.

The permeability coefficients (*P_eff_*) and absorption rate constant (*K_a_*) were obtained for Pur and the Pur-CS/TPP-NPs following in situ perfusion to the different segments. Details of this are presented in Figure 5. For the Pur group, the orders of *P_eff_* and *K_a_* were duodenum > jejunum > ileal > colon, respectively, meaning that the absorption of Pur was the largest in the duodenum, followed by the jejunum, then the ileum, and the absorption of the colon was the weakest. There was a significant difference in the absorption of the duodenum and colon (*p* < 0.05), and there was no significant difference in the absorption between the jejunum and ileum (*p* > 0.05). Compared with the Pur and Pur-CS/TPP-NPs groups, the absorption of the four intestinal segments was significantly increased (*p* < 0.05), and there was also a significant difference between the three low, medium, and high concentrations (*p* < 0.05).

The orders *K_a_* and *P_eff_* of Pur-CS/TPP-NPs prepared by the ionic gelation method were duodenum > jejunum > ileum > colon, and, among them, the absorption of the duodenum, jejunum, and ileum in the low-dose group did not increase significantly, but there was a significant difference (*p* < 0.05), which was significantly different from that of the colon. There was no significant difference between the middle and lower dose groups (*p* > 0.05). There were significant differences between the high-dose and lower-dose groups and the middle-dose group (*p* < 0.05). The *K_a_* and *P_eff_* values of the Pur-CS/TPP-NPs were four times that of Pur, respectively, which is consistent with the pharmacokinetic analysis.

Pur has difficulty penetrating intestinal mucosa due to the steric hindrance caused by glucopyranose on carbon 8. Pur’s optimal intestinal absorption site was the duodenum, which may have a bowel first-pass effect, such that the absorption of Pur into the colon is very low. According to the experimental results, after Pur was made into nanoparticles by CS and TPP, the Pur was wrapped by an ionic gelation method to form nanoparticles. The particle size of the nanoparticles was small, which was beneficial to increasing the absorption area and improving the oral absorption of the drug. This effectively encapsulates Pur and avoids the intestinal first-pass effect, which not only significantly improves Pur absorption in the small intestine but also improves Pur absorption in the colon. There was a significant improvement between different groups, which greatly improved the absorption rate of Pur. The *K_a_* and *P_eff_* values in the Pur-CS/TPP-NPs were four times greater than those in the Pur group, which is consistent with the pharmacokinetic analysis.

## 4. Conclusions

CS and TPP were used as the main materials in this experiment. Pur-CS/TPP-NPs were prepared by an ionic gelation method, with the advantages of being a simple process and presenting no pollution. However, particle size control was the main technical difficulty here. After a single-factor investigation, factors such as the CS concentration, pH of the CS solution, TPP concentration, stirring speed, and stirring time were found to affect the preparation of the Pur-CS/TPP-NPs. The Box–Behnken response surface experiment design was subsequently performed to obtain the optimal Pur-CS/TPP-NPs prescription composition. The prescription verification experiments show that the absolute value of the deviation between the measured value and the predicted value was small, and the optimized nanoparticles were characterized and released in vitro. The results show that the morphologies of the nanoparticles were complete and had a sustained release effect, indicating that the model established here is good. Subsequent studies on pharmacokinetics and intestinal absorption in rats showed that the nanoparticles had a significant, sustained-release effect and significantly improved the bioavailability of Pur and increased bioavailability by a factor of 4.4. The optimization of the Pur nanoparticle formulation process studied in this article, the pharmacokinetic study, and the intestinal absorption research will provide data for the support and referencing of preclinical research for Pur oral nanomedicine. At the same time, the researchers also hope that the data can provide a reference for future research into drugs that are unstable in acids and other drugs similar to puerarin.

## Figures and Tables

**Figure 1 pharmaceutics-12-00216-f001:**
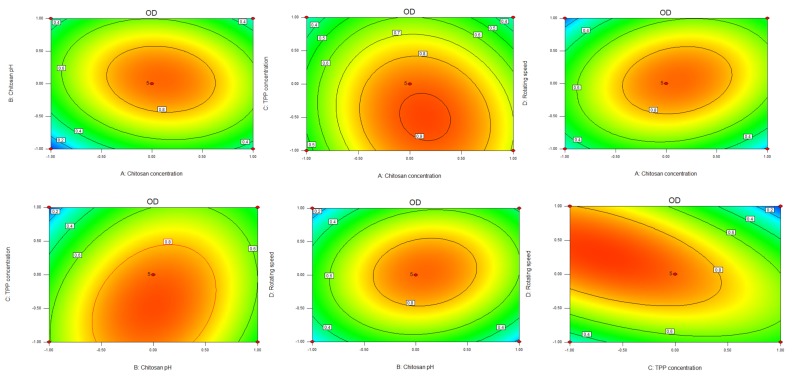
Contour map and 3D surface map of the four factors of effect value OD and A, B, C and D.

**Figure 2 pharmaceutics-12-00216-f002:**
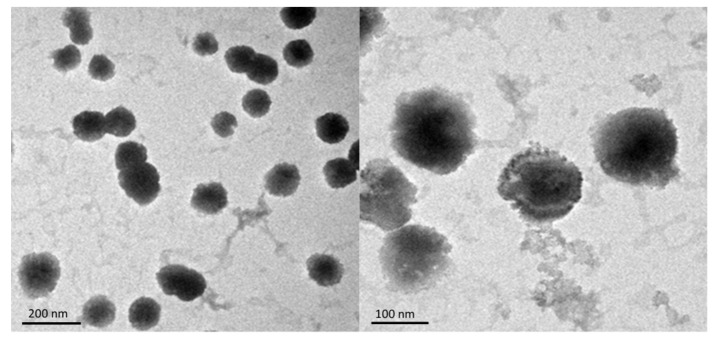
Puerarin chitosan nanoparticles (Pur-CS/TPP-NPs) under transmission electron microscopy.

**Figure 3 pharmaceutics-12-00216-f003:**
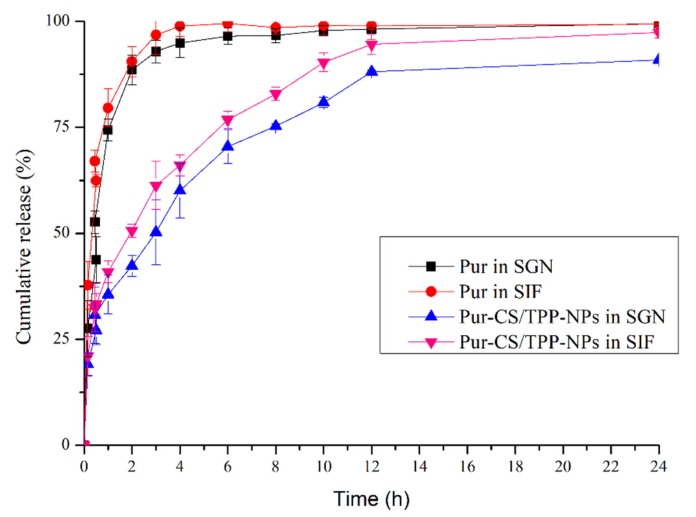
Accumulated release of Pur and Pur-CS/TPP-NPs under artificial gastric juice (SGN) and artificial intestinal fluid (SIF), respectively (*n* = 3, x ± s).

**Figure 4 pharmaceutics-12-00216-f004:**
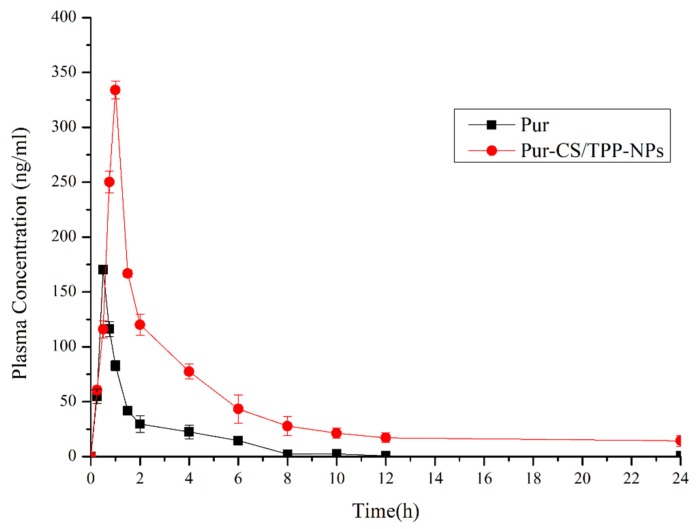
Whole blood concentration versus time profiles of Pur and Pur-CS/TPP-NPs after oral (*n* = 9, x ± s).

**Figure 5 pharmaceutics-12-00216-f005:**
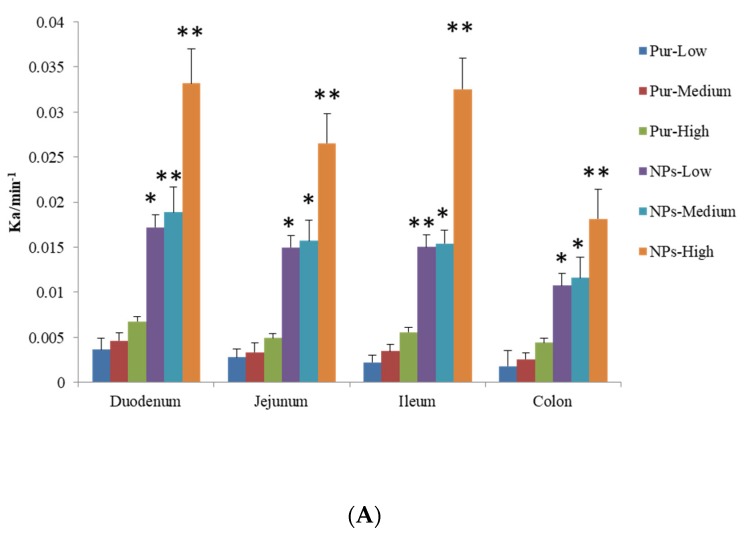
Comparison of *K_a_* (A) and *P_eff_* (B) of Pur and Pur-CS/TPP-NPs in low, medium, and high dose groups. (*n* = 6 x ± s) (* *p* < 0.05, ** *p* < 0.0001 compared with Pur).

**Table 1 pharmaceutics-12-00216-t001:** Box–Behnken response surface optimization experiment factor level table.

Level	A CS Concentration mg/mL	B pH of CS Solution	C TPP Concentration mg/mL	D Speed r/min
−1	0.50	4.50	0.20	300
0	1.25	5.25	0.85	500
1	2.00	6.00	1.50	700

**Table 2 pharmaceutics-12-00216-t002:** Experimental arrangement and results of preparation of puerarin chitosan nanoparticles (Pur-CS/TPP-NPs) Box–Behnken response surface by ion gelation (*n* = 3, x ± s).

Std	Run	A	B	C	D	The Average Particle Size (nm)	PDI	Encapsulation Rate%	Drug Loading%	OD Value
27	1	0	0	0	0	125.07	0.121	92.556	38.565	0.834
21	2	0	−1	0	−1	327.67	0.317	79.810	33.254	0.162
1	3	−1	−1	0	0	349.30	0.337	75.996	31.665	0.000
26	4	0	0	0	0	144.67	0.139	94.280	39.283	0.841
29	5	0	0	0	0	107.83	0.104	97.450	40.604	0.983
23	6	0	−1	0	1	367.60	0.355	76.360	31.817	0.000
3	7	−1	1	0	0	322.70	0.312	78.950	32.896	0.151
17	8	−1	0	−1	0	218.20	0.210	86.796	36.165	0.529
4	9	1	1	0	0	279.30	0.269	79.470	33.113	0.231
22	10	0	1	0	−1	282.30	0.273	80.100	33.375	0.246
25	11	0	0	0	0	134.17	0.129	93.890	39.121	0.851
8	12	0	0	1	1	330.20	0.419	77.904	32.460	0.111
5	13	0	0	−1	−1	337.33	0.326	78.336	32.640	0.111
12	14	1	0	0	1	226.40	0.219	85.680	35.700	0.486
18	15	1	0	−1	0	187.67	0.181	90.140	37.558	0.664
28	16	0	0	0	0	115.30	0.111	94.320	39.300	0.895
6	17	0	0	1	−1	211.97	0.205	83.780	34.908	0.458
16	18	0	1	1	0	224.37	0.217	84.640	35.267	0.463
9	19	−1	0	0	−1	258.90	0.250	83.380	34.742	0.373
19	20	−1	0	1	0	275.33	0.266	81.790	34.079	0.304
14	21	0	1	−1	0	214.87	0.207	87.770	36.571	0.559
2	22	1	−1	0	0	245.80	0.237	82.764	34.485	0.378
7	23	0	0	−1	1	118.73	0.095	87.450	36.438	0.731
15	24	0	−1	1	0	324.90	0.313	78.980	32.908	0.149
10	25	1	0	0	−1	271.90	0.262	80.028	33.345	0.259
11	26	−1	0	0	1	323.53	0.313	79.470	33.113	0.162
24	27	0	1	0	1	254.97	0.246	84.280	35.117	0.402
20	28	1	0	1	0	306.55	0.296	80.100	33.375	0.208
13	29	0	−1	−1	0	184.18	0.178	91.260	38.025	0.696

**Table 3 pharmaceutics-12-00216-t003:** Regression coefficient and analysis of variance (ANOVA) for each factor.

Source	Sum of Squares	Df	Mean of Square	*F* Value	*P*-Value Prob > *F*
Model	2.160	14	0.150	13.010	<0.0001 significant
A-CS concentration	0.042	1	0.042	3.510	0.0819
B-pH of CS solution	0.037	1	0.037	3.130	0.0988
C-TPP concentration	0.210	1	0.210	17.920	0.0008
D-Rotating speed	6.674 × 10^−3^	1	6.674 × 10^−3^	0.560	0.4655
AB	0.022	1	0.022	1.870	0.1928
AC	0.013	1	0.013	1.130	0.3068
AD	0.048	1	0.048	4.040	0.0640
BC	0.051	1	0.051	4.290	0.0573
BD	0.025	1	0.025	2.130	0.1663
CD	0.230	1	0.230	19.710	0.0006
A^2^	0.580	1	0.580	48.790	<0.0001
B^2^	0.740	1	0.740	62.180	<0.0001
C^2^	0.130	1	0.130	11.330	0.0046
D^2^	0.700	1	0.700	59.260	<0.0001
Residual	0.170	14	0.012		
Lack of Fit	0.150	10	0.015	3.940	0.0993 not significant
Pure Error	0.015	4	3.827 × 10^−3^		
Cor Total	2.330	28			

**Table 4 pharmaceutics-12-00216-t004:** Comparison of predicted and measured values (*n* = 3, x ± s).

Comparison Item	Average Particle Size (nm)	PDI	Encapsulation Rate (%)	Drug Loading (%)
Predictive value	125.408	0.121	94.499	39.374
Measured value	126.289 ± 1.175	0.122 ± 0.0005	94.951 ± 0.468	39.596 ± 0.388
RSD	0.930	0.470	0.490	0.980

**Table 5 pharmaceutics-12-00216-t005:** Formula for fitting in vitro release of Pur and Pur-CS/SA-NPs under artificial gastric juice and artificial intestinal juice (*n* = 3, x ± s).

Pur	Model	Simulated Gastric Fluid	R^2^	Artificial Intestinal Juice	R^2^
	Zero-order release	Q = 10.617t + 46.843	0.7386	Q = 0.0865t + 0.6059	0.6607
	Primary release	Ln(1 − Q) = −0.5302t − 0.4971	0.9551	Ln(1 − Q) = −0.8527t − 0.6557	0.9756
	Weibull function	Ln[ln(1/(1 − Q))] = 0.3569t − 0.5345	0.7941	Ln[ln(1/(1 − Q))] = 0.3694t − 0.1545	0.8314
	Higuchi model	Q = 0.3307t(1/2) + 0.2671	0.8816	Q = 0.2753t(1/2) + 0.4342	0.8241
Pur-CS/TPP-NPs	Zero-order release	Q = 0.031t + 0.3757	0.7325	Q = 0.0352t + 0.375	0.6575
	Primary release	Ln(1 − Q) = 0.1934t + 0.1486	0.8433	Ln(1 − Q) = −0.1369t − 0.4358	0.8726
	Weibull function	Ln[ln(1/(1 − Q))] = 0.1646t − 1.3747	0.7053	Ln[ln(1/(1 − Q))] = 0.0945t − 0.5492	0.6986
	Higuchi model	Q = −0.1044t(1/2) − 0.3765	0.8802	Q = 0.1989 t(1/2) + 0.1894	0.9007

**Table 6 pharmaceutics-12-00216-t006:** Pharmacokinetic parameters of Pur and Pur-CS/TPP-NPs.

Parameter	Pur	Pur-CS/TPP-NPs
AUC _0–24_(mg/L*h)	250.087 ± 32.156	1108.583 ± 234.534 *
AUC _0–∞_(mg/L*h)	250.091 ± 28.398	1182.926 ± 265.746 *
AUMC _0–24_	702.234 ± 87.986	7302.349 ± 1175.975 *
AUMC _0–∞_	742.336 ± 93.231	14,653.559 ± 4634.864 *
MRT _0–24_(h)	2.808 ± 0.014	6.587 ± 0.132
MRT _0–∞_(h)	2.968 ± 0.121	12.388 ± 3.430 *
VRT _0–24_(h^2^)	14.691 ± 2.146	49.163 ± 7.768
VRT _0–∞_(h^2^)	17.971 ± 1.975	171.732 ± 20.325 *
MAT(h)	2.611 ± 0.532	11.199 ± 2.546
T_max_(h)	0.500 ± 0.031	1.000 ± 0.076
C_max_(mg/L)	191.830 ± 17.963	343.314 ± 56.760
CL/F(L/h/kg)	0.991 ± 0.021	0.154 ± 0.011

* *p* < 0.05, compared with Pur.

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
