# Peer review of "Preparation of Puerarin Chitosan Oral Nanoparticles by Ionic Gelation Method and Its Related Kinetics"

_pharmaceutics, 2020, doi:10.3390/pharmaceutics12030216_

Round 1

Reviewer 1 Report

Summary: Accept after minor revision.

Yan et al. demonstrate the formation of puerarin chitosan nanoparticles. The involved parameters were optimized, and the efficacy of this best formulation was validated using release studies and intestinal perfusion. These findings echo a well known result that nanoparticles can improve the efficiency of delivery. Overall, the study is well executed, and this manuscript is well suited for Pharmaceutics.

Positives:

Demonstrates the efficacy of nanoparticles over free administration of drugs. Optimization of 4 key parameters followed by validation. Determination of drug plasma concentration and puerarin absorption across different sections of the intestine.

Critique: None noted, the experimental design is appropriate for the submitted journal. Moderate English check and formatting required before publication.

Author Response

Dear Reviewer:

  Thank you for the reviewers’ comments concerning our manuscript entitled “Preparation of Puerarin chitosan oral nanoparticles by ionic gelation method and its related kinetics”. Those comments are all valuable and very helpful for revising and improving our paper, as well as the important guiding significance to our researches. We have studied comments carefully and have made correction which we hope meet with approval. The main corrections in the paper and the responds to the reviewer’s comments are as following:

Thank you very much for your recognition of our work for demonstrates the efficacy of nanoparticles over free administration of drugs. Optimization of 4 keys parameters followed by validation. Determination of drug plasma concentration and puerarin absorption across different sections of the intestine. We will continue to study this topic in the future.

For "None noted, the experimental design is appropriate for the submitted journal. Moderate English check and formatting required before publication". This experimental design was appropriate for Pharmaceutics and we had moderate English checking and formatting.

Finally, thank you very much for your comments.

Reviewer 2 Report

A manuscript entitled "Preparation of Puerarin chitosan oral nanoparticles by ionic gelation method and its related kinetics" by Jie Yan et al. is contained some interesting findings. However, the current content of this article doesn't reach the level of this Journal. In fact, there are many manuscripts of CS/TPP-NPs as drug carriers. Further, except for chitosan, there are many manuscripts of improvement of bioavailability of Puerarin using some drug carriers such as nanocrystal, microemulsion, cyclodextrin. The authors should be clear the significant novelty of Pur-CS/TPP-NPs, compared with these materials. This paper does not reach the level of this journal. As minor points, there are some mistypes, a lack of some abbreviations, the misorder of figure&tables. The authors should correct these parts.

Author Response

Response to Reviewer 2 Comments

Dear Reviewer:

Thank you for the reviewers’ comments concerning our manuscript entitled “Preparation of Puerarin chitosan oral nanoparticles by ionic gelation method and its related kinetics”. Those comments are all valuable and very helpful for revising and improving our paper, as well as the important guiding significance to our researches. We have studied comments carefully and have made correction which we hope meet with approval. The main corrections in the paper and the responds to the reviewer’s comments are as following:

Point 1: There are many manuscripts of CS/TPP-NPs as drug carriers. Further, except for chitosan, there are many manuscripts of improvement of bioavailability of Puerarin using some drug carriers such as nanocrystal, microemulsion, cyclodextrin. The authors should be clear the significant novelty of Pur-CS/TPP-NPs, compared with these materials.  This paper does not reach the level of this journal.

Response 1: The chitosan used in this experiment had high degree of deacetylation (99%). Compared to other types of chitosan, It could be dissolved without acid, and could be dissolved with ultrapure water directly, and the preparation process of nanoparticles was simple. It also had good water-solubility, stability and film-forming properties. Further more, the chitosan we used, expands the drug application range of the nanoparticle preparation method by ion gelation. Instability drugs in acid could also be prepared by this method, which makeed many drugs break through the restrictions.

Compared with nanocrystal, microemulsion, cyclodextrin, The particle size and PDI of the nanoparticles prepared by the water-soluble chitosan drug-loaded by ionic gelation method were stable, and the encapsulation rate was more than 90%, no sudden release of nanoparticles and the bioavailability of puerarin was significantly improved. In addition, this method has simple operation, mild reaction conditions, and was easy to form industrialization.

Point 2: As minor points, there are some mistypes, a lack of some abbreviations, the misorder of figure&tables. The authors should correct these parts.

Response 2: Some mistypes, a lack of some abbreviations, the misorder of figure &tables had been modified.

Finally, thank you very much for your comments.

Reviewer 3 Report

The paper is devoted to the formulation of puerarin nanoparticles for oral delivery. The techniques employed are adequate and the results are appropriately discussed and support the conclusions attained. Nevertheless, an experimental design for optimizing the variables tested would have been more appropriate if optimization is required. 

1. Purerarin is a BCS IV drug with low oral bioavailability. Optimization of component and composition is needed to be further explain. In addition, it is important to describe how CS/TPP-NP increases the release and absorption of puerarin.

2. Statistical analysis of the results in Fig. 3, 4, and Table 6 is missing. Moreover, the figure legend for Fig. 5 is also missing. For Fig. 5, statistical anlaysis is decribed in the manuscript, but significant differences are not presented in Figure.

3. In p. 17, please check "CSpH."

Author Response

Response to Reviewer 3 Comments

Dear Reviewer:

  Thank you for the reviewers’ comments concerning our manuscript entitled “Preparation of Puerarin chitosan oral nanoparticles by ionic gelation method and its related kinetics”. Those comments are all valuable and very helpful for revising and improving our paper, as well as the important guiding significance to our researches. We have studied comments carefully and have made correction which we hope meet with approval. The main corrections in the paper and the responds to the reviewer’s comments are as following:

Before the process optimization in this article, the research team first went through a detailed single-factor inspection process, and examined the factors that may affect the particle size, PDI, and encapsulation efficiency of drug loading.The concentration of chitosan, the pH of the chitosan solution, the concentration of sodium tripolyphosphate, the stirring time, the stirring speed, the temperature, etc. were examined separately. The specific investigation was arranged in the "3.3.4" part of the article. The results are analyzed in the "2.1.1" section of the article. After a series of investigations, the pH value of the chitosan solution, the concentration of chitosan, the concentration of sodium tripolyphosphate, and the stirring speed were the four factors that affected the nanoparticles to optimize the Box-Behnken effect surface. And then get the best prescription of nanoparticles, and perform parallel experiments on the best prescription to determine the stability of the prescription. In view of the simpler and more space-consuming process of single-factor investigation, this article briefly introduces the single-factor investigation process and results in the "2.1.1" part of the body.

  Point 1: Purerarin is a BCS IV drug with low oral bioavailability. Optimization of component and composition is needed to be further explain. In addition, it is important to describe how CS/TPP-NP increases the release and absorption of puerarin.

Response 1:The main problem of puerarin was low bioavailability, and puerarin was a monomer compound that was difficult to obtain by chemical synthesis, and the commercial exploitation of natural resources was also relatively limited. Therefore, we improved the oral bioavailability by making nanoparticles, and the results showed that the relative bioavailability of puerarin was increased by 4.4 times. On the one hand, it could save resources and protect the ecology. On the other hand, the use of puerarin in the treatment of cardiovascular and cerebrovascular diseases, liver diseases and other diseases in Chinese folks had great clinical needed and great market potential. Therefore, the research in this experiment had great value for the future development of puerarin.

Point 2: How CS/TPP-NP increases the release and absorption of puerarin.

Response 2: Spontaneous reaction combination of cation of chitosan and anion of sodium tripolyphosphate, forming a polyelectrolyte complex named TPP/Chit. This complex is stabilized by cross-linked electrostatic interaction between chitosan NH3+ and TPP-O- groups, resulting in a three-dimensional entanglement that precipitates from an aqueous solution in the form of gel-like nanoparticles, also named microgels and then ncapsulation of puerarin to form nanoparticles.

Point 3: Statistical analysis of the results in Fig. 3, 4, and Table 6 is missing. Moreover, the figure legend for Fig. 5 is also missing. For Fig. 5, statistical anlaysis is decribed in the manuscript, but significant differences are not presented in Figure.    

Response 3:The missing content in the article had been completed and checked correctly, and the format of the figure had been modified accordingly to make up the significance.

Point 4:Please check "CSpH."

Response 4: "CSpH" means the pH of chitosan solution, had been modified

Finally, thank you very much for your comments.

Reviewer 4 Report

The paper presented very confusing results, lacks on methodology and, previously to be considered for publication there are several major points to be considered and solved:

Writing errors should be revised. In fact, the manuscript needs a deep language review and edition.

Introduction:

Abbreviators should be revised and defined correctly; The aim of this study is not presented clearly in this section;

Results and discussion section:

In some situations, no information about statistical analysis is mentioned, neither the replicates nor independent experiments considered in the results presented. There are several missing points and data, namely in section 2.1.1 (lines74-90), and contraditory data considering the methodology described in the following section (e.g. particle size considering the “extrusion” process by 0.22 filter) These sections should be improved with information and critical analysis of results obtained. The relevance of results obtained and the differences obtained for the different polymer concentration, pH, etc. are not discussed or analyzed.

In general, results are presented without information of data analysis or an eventual statistical analysis. The manuscript will benefit also with a more objective presentation of results and with a discussion that can highlight the real improvements and contributes of this work for the knowledge in this area comparing to other published works.

Materials and Method section:

This section lacks the description of materials used (e.g. no information about the TPP used, or the identification of chitosan (molecular weight, provider, cas, etc.). These details are relevant for the analysis of the work presented, due to its relevance for the bioactivity and Physico-chemical behaviour. How and why it was selected the Pur concentration? it should be justified considering the claimed clinical application and commercial formulations available; In vitro biocompatibility assays were not performed or described previously to in vivo assays. Despite the very important and adequate pharmacokinetic and release control assays performed in different artificial physiological fluids, these in vitro assays should be considered previously to in vivo assays and according to international guidelines for animal experimentation; Furthermore, and considering the ARRIVE Guidelines Checklist with respect to Reporting In Vivo Experiments the manuscript lacks on the methodology description, once were not identified the ethical concerns, experimental conditions, acclimatization procedures, groupings of animals, control group, etc.

The conclusion section should be reinforced highlighting the relevance and contribution of this study to clinical practice.

Author Response

 Dear Reviewer:

Thank you for the reviewers’ comments concerning our manuscript entitled “Preparation of Puerarin chitosan oral nanoparticles by ionic gelation method and its related kinetics”. Those comments are all valuable and very helpful for revising and improving our paper, as well as the important guiding significance to our researches. We have studied comments carefully and have made correction which we hope meet with approval. The main corrections in the paper and the responds to the reviewer’s comments are as following:

Point 1: The paper presented very confusing results, lacks on methodology. Abbreviators should be revised and defined correctly; The aim of this study is not presented clearly in this section.

Response 1: Missing methodologies had been completed in the text. The abbreviators had checked, upon request defined in parentheses the first time they appear in the abstract, main text, and in figure or table captions and used consistently thereafter. The purpose of this article had been proposed in the introduction. In this paper, puerarin oral nanoparticles were prepared by ionic crosslinking method. With a view to lay a certain foundation for puerarin in the future oral drug development, clinical application and industrial development. At the same time the material expands the drug application range of the nanoparticle preparation method by ion gelation. Instability drugs in acid can also be prepared by this method, which makes many drugs break through the restrictions.

Point 2: No information about statistical analysis is mentioned, neither the replicates nor independent experiments considered in the results presented .

Response 2: "3.3.8.5" mentioned in the article related to statistical analysis of the data, the rest of the data were based on mean ± standard deviation.

Point 3: There are several missing points and data, namely in section 2.1.1 (lines74-90), and contraditory data considering the methodology described in the following section (e.g. particle size considering the “extrusion” process by 0.22 filter) These sections should be improved with information and critical analysis of results obtained. The relevance of results obtained and the differences obtained for the different polymer concentration, pH, etc. are not discussed or analyzed.

Response 3: Relevant data had been checked, and missing parts had been completed. Particle size considering the “extrusion” process by particle size considering the “extrusion” process by 0.22 filter, The verification showed that the 0.22 filter had no effect on the particle size, PDI, encapsulation efficiency, and drug loading of the nanoparticles. The 0.22 filter was passed here to prevent impurities from affecting the experiment. This experiment carefully examined the concentration of chitosan and the pH value to check the concentration of chitosan solution, sodium tripolyphosphate concentration, stirring time, stirring speed, temperature, etc. The effect of the amount was shown in the "2.1.1" section of this article.

Point 4: This section lacks the description of materials used (e.g. no information about the TPP used, or the identification of chitosan (molecular weight, provider, cas, etc.). These details are relevant for the analysis of the work presented, due to its relevance for the bioactivity and Physico-chemical behavior.

Response 4: The chitosan used in this experiment was made in the laboratory, and because of its high degree of deacetylation, it had more water-solubility characteristics than other types of chitosan, so this experiment used this chitosan as a carrier material to prepare nanoparticles. At present, chitosan used in this experiment could be obtained through donation. Sodium Tripolyphosphate was purchased from Shanghai Yien Chemical Technology Co., Ltd.(Beijing. China).

Point 5: How and why it was selected the Pur concentration? it should be justified considering the claimed clinical application and commercial formulations available; In vitro biocompatibility assays were not performed or described previously to in vivo assays. Despite the very important and adequate pharmacokinetic and release control assays performed in different artificial physiological fluids, these in vitro assays should be considered previously to in vivo assays and according to international guidelines for animal experimentation.

Response 5: In the early stage of the experiment, we found that when the puerarin concentration was too high, crystals would precipitate, making the prepared nanoparticles larger in particle size, and the PDI was also very large. Therefore, the optimal puerarin concentration was selected in the experiment for corresponding investigation. And we had considered previously to in vivo assays and according to international guidelines for animal experimentation. The material part of the article had been modified as required to correct animal ethical requirements.

Point 6: The conclusion section should be reinforced highlighting the relevance and contribution of this study to clinical practice.

Response 6: In the results section of the article, a discussion of the corresponding results had been added.

Finally, thank you very much for your comments.

Round 2

Reviewer 2 Report

 Authors thoughtfully revised manuscript and addressed reviewer comments by adding additional figures. I think this manuscript is suitable for publication in Pharmaceutics.

Author Response

Dear Reviewer:

 Thanks to your reviewers for commenting on our manuscript entitled " Preparation of Puerarin chitosan oral nanoparticles by ionic gelation method and its related kinetics ".

Thank you very much for your recognition of our work. Your recognition is the driving force for our progress, and we will continue to study this topic in the future.

Finally, thank you very much for your comments and suggestions.

Kind regards

Reviewer 3 Report

Please explain the optimization of nanoparticle formulation to improve the oral bioavailability of puerarin.

Author Response

Dear Reviewer:

  Thanks to your reviewers for commenting on our manuscript entitled "Preparation of Puerarin chitosan oral nanoparticles by ionic gelation method and its related kinetics".

First of all, thank you very much for your recognition of our work. Your recognition is our driving force for progress, and we will continue to study this topic in the future.

Point 1: Please explain the optimization of nanoparticle formulation to improve the oral bioavailability of puerarin.

Response 1: In this paper, we investigated the concentration of chitosan, the pH value of chitosan, the concentration of TPP solution, the stirring speed, and the stirring time through single-factor investigations. The particle size, PDI, encapsulation efficiency, and drug loading were used as the indicators of investigation. Finally, four factors with the greatest influence were selected, namely chitosan concentration, chitosan pH value, TPP solution concentration, stirring speed, and optimization of the Box-Behnken effect surface to obtain the best optimized prescription. The inspection indicators selected in this article were particle size, PDI, encapsulation efficiency, and drug loading. Because in the process of preparing nanoparticles, the particle size was the most important factor for the preparation of nanoparticles, and it was also the most difficult to control. The particle size was too large to be easily absorbed. By examining the particle size, the smaller the particle size in a certain range, the better, and finally it was stable at 120 nm, which was most conducive to absorption. The smaller the PDI, the more uniform the prepared nanoparticles, so that the nanoparticles would not be large or small. At the same time, the encapsulation rate and drug loading capacity were taken as indicators to maximize the encapsulation efficiency and drug loading capacity of the nanoparticles, maximize the embedding amount of puerarin, and improve the absorption of puerarin. By examining these indicators to optimize the prescription design of nanoparticles, the bioavailability of puerarin could be significantly improved.

Finally, thank you very much for your comments and suggestions.

Kind regards

Reviewer 4 Report

I think authors of the manuscript did a good job. However, I still concern about the methodological approach and discussion (2 references in this section) that should highlight the real improvements and contributes of the work (critical analysis of results obtained) and should revised results obtained and compare with other published works.

In my opinion, and considering this critical point, the revised manuscript still not suitable for publication.

Author Response

Dear Reviewer:

  Thanks to your reviewers for commenting on our manuscript entitled “Preparation of Puerarin chitosan oral nanoparticles by ionic gelation method and its related kinetics”.

Point 1: I think authors of the manuscript did a good job. However, I still concern about the methodological approach and discussion (2 references in this section) that should highlight the real improvements and contributes of the work (critical analysis of results obtained) and should revised results obtained and compare with other published works.

Response 1: First of all, thank you very much for your recognition of our work. Your recognition is our driving force for progress, and we will continue to study this topic in the future.

In this paper, water-soluble chitosan and non-toxic TPP were used to prepare puerarin chitosan oral nanoparticles in ionic cross-linked form, which greatly improved the oral bioavailability of puerarin. Puerarin is currently studying solid lipid nanoparticles. Solid lipid nanoparticles were more suitable for lipophilic drugs. However, puerarin could form intermolecular hydrogen bonds due to having two phenolic hydroxyl groups at the 7 and 4' positions, and the molecules were closely packed, which increased the intermolecular force of puerarin, it had a high melting point, and poor fat solubility, water solubility. Solid lipid nanoparticles had low encapsulation efficiency and drug loading capacity, and had sudden release. Compared with nanocrystal, microemulsion, cyclodextrin, the particle size and PDI of the nanoparticles prepared by the water-soluble chitosan drug-loaded by ionic gelation method were stable, and the encapsulation rate was more than 90%, no sudden release of nanoparticles and the bioavailability of puerarin was significantly improved.

Ionic gelation method had simple preparation process, mild reaction conditions, non-toxic, and very easy to control. The most important advantage was that the most important influencing factor of nanoparticles was the control of particle size. This method could achieve a good effect of controlling the particle size, and the prepared nanoparticles had a uniform particle size distribution, and at the same time, the encapsulation efficiency and drug loading were relatively high.

In the research of puerarin, the preparation of nanoparticles by ionic gelation had not been studied in depth, so this article had conducted a thorough and comprehensive study on the preparation of puerarin chitosan oral nanoparticles by ionic gelation.

In this paper, the preparation of oral puerarin chitosan nanoparticles by ionic gelation method was systematically investigated. In the early stage, the factors that had the greatest influence on the prescription were screened through single-factor investigation, and then the Box-Behnken effect surface was optimized. Box-Behnken was considered way more efficient and economical, than the corresponding three-level factorial design and had been extensively utilized for the optimization of RME due to the lower number of experiments required (Sy Mohamad Sharifah Fathiyah,Mohd Said Farhan,Abdul Munaim Mimi Sakinah et al. Application of experimental designs and response surface methods in screening and optimization of reverse micellar extraction.[J] .Crit. Rev. Biotechnol., 2020, undefined: 1-16.). The optimized prescription was subjected to multiple verification tests to confirm the feasibility of the prescription. The nanoparticles were then subjected to a series of in vitro characterizations as well as in vivo pharmacokinetics and rat intestinal absorption studies in vivo. Due to the steric hindrance caused by glucopyranose on carbon 8, puerarin was difficult to penetrate into the intestinal mucosa. puerarin's optimal intestinal absorption site was the duodenum, which may had a bowel first-pass effect. According to the experimental results, after puerarin was made into nanoparticles with chitosan and sodium tripolyphosphate, puerarin was encapsulated to form nanoparticles by the ionic gel method, which greatly improved the absorption rate of puerarin. The results showed that the prepared nanoparticles significantly increased the absorption of puerarin.

We hope that through this article, it will lay the foundation for the future research of puerarin nano preparations and the future industrialization development.

Finally, thank you very much for your comments and suggestions.

Kind regards

Round 3

Reviewer 4 Report

Authors of the manuscript did a good job and improved or justified different questions.

Author Response

Dear Reviewer:

Thanks to your reviewers for commenting on our manuscript entitled " Preparation of Puerarin chitosan oral nanoparticles by ionic gelation method and its related kinetics ". Thank you very much for your recognition of our work. Your recognition is the driving force for our progress, and we will continue to study this topic in the future.

In the first two rounds of replies, because we responded a lot, the expression was not very clear. Although our method during the research was not very innovative, we carried out a systematic and detailed study of the nanoparticles prepared by this method.

  1. The materials we used were special and had not been used by other experimenters. The chitosan used in this experiment had high degree of deacetylation. It could be dissolved without acid, and could be dissolved with ultrapure water directly. It also had good water-solubility, stability and film-forming properties, the preparation process of nanoparticles was simple.
  2. For puerarin itself, puerarin could form intermolecular hydrogen bonds due to had two phenolic hydroxyl groups at the 7 and 4' positions, and the molecules were closely packed, which increased the intermolecular force of puerarin, it had a high melting point, and poor fat solubility and water solubility. We hoped to use puerarin as a model drug. This method could be applied to all drugs similar the properties of puerarin.
  3. In the early stage of the experiment, we did a lot of literature research. Through the literature we found that, at present, more studied on improving the bioavailability of puerarin were structural modification. This method was complicated to operate and easily destroyed the activity of puerarin itself. In addition, there were also studied on microemulsions, phospholipid complexes, and nanoparticles, of which solid lipid nanoparticles had been the most studied. However, in view of the nature of puerarin, the encapsulation efficiency and drug loading capacity of solid lipid nano were not high. Compared with nanocrystal, microemulsion, cyclodextrin, the particle size and PDI of the nanoparticles prepared by the water-soluble chitosan drug-loaded by ionic gelation method were stable, and the encapsulation rate was more than 90%, no sudden release of nanoparticles and the bioavailability of puerarin was significantly improved. The ionic gelation method used in this paper had simple preparation process, mild reaction conditions, non-toxicity, and easy control. The particle size of nanoparticles was an important factor affecting absorption, and the biggest advantage of this method was that it was easy to control the particle size of nanoparticles. This method prepared nanoparticles had a uniform particle size distribution, and the encapsulation efficiency and drug loading were high.

Finally, thank you very much for your comments and suggestions.

Kind regards
